# Efficient Modeling of Latent Information in Supervised Learning using Gaussian Processes

**Zhenwen Dai** [*‡]
zhenwend@amazon.com

**Mauricio A. Álvarez** [†]
mauricio.alvarez@sheffield.ac.uk

**Neil D. Lawrence** [†‡]
lawrennd@amazon.com

## Abstract

Often in machine learning, data are collected as a combination of multiple conditions, e.g., the voice recordings of multiple persons, each labeled with an ID. How could we build a model that captures the latent information related to these conditions and generalize to a new one with few data? We present a new model called *Latent Variable Multiple Output Gaussian Processes (LVMOGP)* that allows to jointly model multiple conditions for regression and generalize to a new condition with a few data points at test time. LVMOGP infers the posteriors of Gaussian processes together with a latent space representing the information about different conditions. We derive an efficient variational inference method for LVMOGP for which the computational complexity is as low as sparse Gaussian processes. We show that LVMOGP significantly outperforms related Gaussian process methods on various tasks with both synthetic and real data.

## 1   Introduction

Machine learning has been very successful in providing tools for learning a function mapping from an input to an output, which is typically referred to as supervised learning. One of the most pronouncing examples currently is deep neural networks (DNN), which empowers a wide range of applications such as computer vision, speech recognition, natural language processing and machine translation [Krizhevsky et al., 2012, Sutskever et al., 2014]. The modeling in terms of function mapping assumes a one/many to one mapping between input and output. In other words, ideally the input should contain sufficient information to uniquely determine the output apart from some sensory noise.

Unfortunately, in most cases, this assumption does not hold. We often collect data as a combination of multiple scenarios, e.g., the voice recording of multiple persons, the images taken from different models of cameras. We only have some labels to identify these scenarios in our data, e.g., we can have the names of the speakers and the specifications of the used cameras. These labels themselves do not represent the full information about these scenarios. A question therefore is how to use these labels in a supervised learning task. A common practice in this case would be to ignore the difference of scenarios, but this will result in low accuracy of modeling, because all the variations related to the different scenarios are considered as the observation noise, as different scenarios are not distinguishable anymore in the inputs,. Alternatively, we can either model each scenario separately, which often suffers from too small training data, or use a one-hot encoding to represent each scenario. In both of these cases, generalization/transfer to new scenario is not possible.

---

[*]Inferentia Limited.

[†]Dept. of Computer Science, University of Sheffield, Sheffield, UK.

[‡]Amazon.com. The scientific idea and a preliminary version of code were developed prior to joining Amazon.

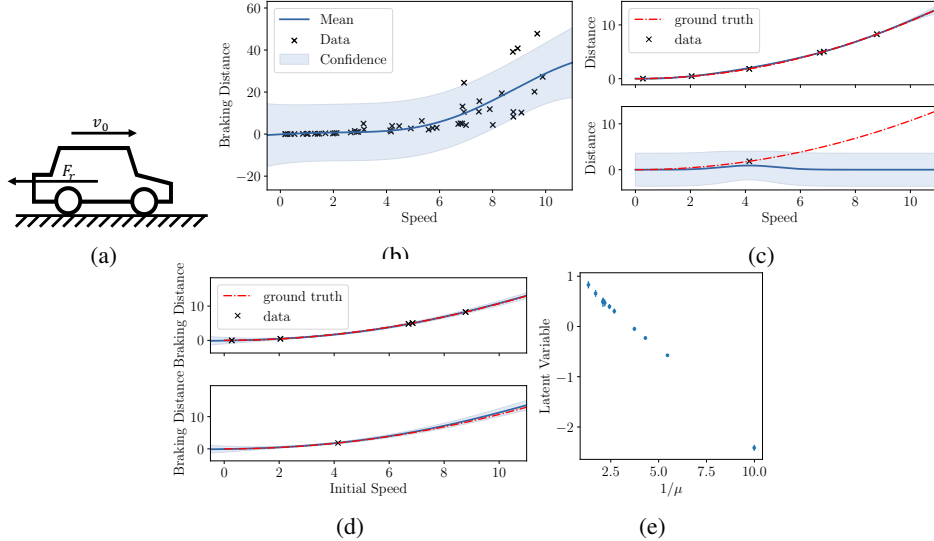

(a)     (b)     (c)

(d)     (e)

Figure 1: A toy example about modeling the braking distance of a car. (a) illustrating a car with the initial speed $v_0$ on a flat road starts to brake due to the friction force $F_r$. (b) the results of a GP regression on all the data from 10 different road and tyre conditions. (c) The top plot visualizes the fitted model with respect to one of the conditions in the training data and the bottom plot shows the prediction of the trained model for a new condition with only one observation. The model assumes every condition independently. (d) LVMOGP captures the correlation among different conditions and the plot shows the curve with respect to one of the conditions. By using the information from all the conditions, it is able to predict in a new condition with only one observation.(e) The learned latent variable with uncertainty corresponds to a linear transformation of the inverse of the true friction coefficient ($\mu$). The blue error bars denote the variational posterior of the latent variables $q(\mathbf{H})$.

In this paper, we address this problem by proposing a probabilistic model that can jointly consider different scenarios and enables efficient generalization to new scenarios. Our model is based on Gaussian Processes (GP) augmented with additional latent variables. The model is able to represent the data variance related to different scenarios in the latent space, where each location corresponds to a different scenario. When encountering a new scenario, the model is able to efficient infer the posterior distribution of the location of the new scenario in the latent space. This allows the model to efficiently and robustly generalize to a new scenario. An efficient Bayesian inference method of the propose model is developed by deriving a closed-form variational lower bound for the model. Additionally, with assuming a Kronecker product structure in the variational posterior, the derived stochastic variational inference method achieves the same computational complexity as a typical sparse Gaussian process model with independent output dimensions.

## 2 Modeling Latent Information

### 2.1 A Toy Problem

Let us consider a toy example where we wish to model the braking distance of a car in a completely data-driven way. Assuming that we do not know physics about car, we could treat it as a non-parametric regression problem, where the input is the initial speed read from the speedometer and the output is the distance from the location where the car starts to brake to the point where the car is fully stopped. We know that the braking distance depends on the friction coefficient, which varies according to the condition of the tyres and road. As the friction coefficient is difficult to measure directly, we can conduct experiments with a set of different tyre and road conditions, each associated with a condition id, e.g., ten different conditions, each has five experiments with different initial speeds. *How can we model the relation between the speed and distance in a data-driven way, so that we can extrapolate to a new condition with only one experiment?*

Denote the speed to be $x$, the observed braking distance to be $y$, and the condition id to be $d$. A straight-forward modeling choice is to ignore the difference in conditions. Then, the relation between

the speed and the distance can be modeled as

$$y = f(x) + \epsilon, \quad f \sim GP, \tag{1}$$

where $\epsilon$ represents measurement noise, and the function $f$ is modeled as a Gaussian Process (GP). Since we do not know the parametric form of the function, we model it non-parametrically. The drawback of this model is that the accuracy is very low as all the variations caused by different conditions are modeled as measurement noise (see Figure 1b). Alternatively, we can model each condition separately, i.e., $f_d \sim GP, d = 1, \ldots, D$, where $D$ denotes the number of considered conditions. In this case, the relation between speed and distance for each condition can be modeled cleanly if there are sufficient data in that condition. However, such modeling is not able to generalize to new conditions (see Figure 1c), because it does not consider the correlations among conditions.

Ideally, we wish to model the relation together with the *latent information* associated with different conditions, i.e., the friction coefficient in this example. A probabilistic approach is to assume a latent variable. With a latent variable $\mathbf{h}_d$ that represents the latent information associated with the condition $d$, the relation between speed and distance for the condition $d$ is, then, modeled as

$$y = f(x, \mathbf{h}_d) + \epsilon, \quad f \sim GP, \quad \mathbf{h}_d \sim \mathcal{N}(0, \mathbf{I}). \tag{2}$$

Note that the function $f$ is shared across all the conditions like in (1), while for each condition a different latent variable $\mathbf{h}_d$ is inferred. As all the conditions are jointly modeled, the correlation among different conditions are correctly captured, which enables generalization to new conditions (see Figure 1d for the results of the proposed model).

This model enables us to capture the relation between the speed, distance as well as the latent information. The latent information is learned into a latent space, where each condition is encoded as a point in the latent space. Figure 1e shows how the model "discovers" the concept of friction coefficient by learning the latent variable as a linear transformation of the inverse of the true friction coefficients. With this latent representation, we are able to infer the posterior distribution of a new condition given only one observation and it gives reasonable prediction for the speed-distance relation with uncertainty.

## 2.2 Latent Variable Multiple Output Gaussian Processes

In general, we denote the set of inputs as $\mathbf{X} = [\mathbf{x}_1, \ldots, \mathbf{x}_N]^\top$, which corresponds to the speed in the toy example, and each input $\mathbf{x}_n$ can be considered in $D$ different conditions in the training data. For simplicity, we assume that, given an input $\mathbf{x}_n$, the outputs associated with all the $D$ conditions are observed, denoted as $\mathbf{y}_n = [y_{n1}, \ldots, y_{nD}]^\top$ and $\mathbf{Y} = [\mathbf{y}_1, \ldots, \mathbf{y}_N]^\top$. The latent variables representing different conditions are denoted as $\mathbf{H} = [\mathbf{h}_1, \ldots, \mathbf{h}_D]^\top, \mathbf{h}_d \in \mathbb{R}^{Q_H}$. The dimensionality of the latent space $Q_H$ needs to be pre-specified like in other latent variable models. The more general case where each condition has a different set of inputs and outputs will be discussed in Section 4.

Unfortunately, inference of the model in (2) is challenging, because the integral for computing the marginal likelihood, $p(\mathbf{Y}|\mathbf{X}) = \int p(\mathbf{Y}|\mathbf{X}, \mathbf{H})p(\mathbf{H})\mathrm{d}\mathbf{H}$, is analytically intractable. Apart from the analytical intractability, the computation of the likelihood $p(\mathbf{Y}|\mathbf{X}, \mathbf{H})$ is also very expensive, because of its cubic complexity $O((ND)^3)$. To enable efficient inference, we propose a new model which assumes the covariance matrix can be decomposed as a Kronecker product of the covariance matrix of the latent variables $\mathbf{K}^H$ and the covariance matrix of the inputs $\mathbf{K}^X$. We call the new model Latent Variable Multiple Output Gaussian Processes (LVMOGP) due to its connection with multiple output Gaussian processes. The probabilistic distributions of LVMOGP are defined as

$$p(\mathbf{Y}_:|\mathbf{F}_:) = \mathcal{N}\left(\mathbf{Y}_:|\mathbf{F}_:, \sigma^2 \mathbf{I}\right), \quad p(\mathbf{F}_:|\mathbf{X}, \mathbf{H}) = \mathcal{N}\left(\mathbf{F}_:|0, \mathbf{K}^H \otimes \mathbf{K}^X\right), \tag{3}$$

where the latent variables $\mathbf{H}$ have unit Gaussian priors, $\mathbf{h}_d \sim \mathcal{N}(0, \mathbf{I})$, $\mathbf{F} = [\mathbf{f}_1, \ldots, \mathbf{f}_N]^\top, \mathbf{f}_n \in \mathbb{R}^D$ denote the noise-free observations, the notation ":" represents the vectorization of a matrix, e.g., $\mathbf{Y}_: = \mathrm{vec}(\mathbf{Y})$ and $\otimes$ denotes the Kronecker product. $\mathbf{K}^X$ denotes the covariance matrix computed on the inputs $\mathbf{X}$ with the kernel function $k_X$ and $\mathbf{K}^H$ denotes the covariance matrix computed on the latent variable $\mathbf{H}$ with the kernel function $k_H$. Note that the definition of LVMOGP only introduces a Kronecker product structure in the kernel, which does not directly avoid the intractability of its marginal likelihood. In the next section, we will show how the Kronecker product structure can be used for deriving an efficient variational lower bound.

# 3  Scalable Variational Inference

The exact inference of LVMOGP in (3) is analytically intractable due to an integral of the latent variable in the marginal likelihood. Titsias and Lawrence [2010] develop a variational inference method by deriving a closed-form variational lower bound for a Gaussian process model with latent variables, known as Bayesian Gaussian process latent variable model. Their method is applicable to a broad family of models including the one in (2), but is not efficient for LVMOGP because it has cubic complexity with respect to $D$.[4] In this section, we derive a variational lower bound that has the same complexity as a sparse Gaussian process assuming independent outputs by exploiting the Kronecker product structure of the kernel of LVMOGP.

We augment the model with an auxiliary variable, known as the *inducing variable* $\mathbf{U}$, following the same Gaussian process prior $p(\mathbf{U}_:) = \mathcal{N}\left(\mathbf{U}_:|0, \mathbf{K}_{uu}\right)$. The covariance matrix $\mathbf{K}_{uu}$ is defined as $\mathbf{K}_{uu} = \mathbf{K}_{uu}^H \otimes \mathbf{K}_{uu}^X$ following the assumption of the Kronecker product decomposition in (3), where $\mathbf{K}_{uu}^H$ is computed on a set of *inducing inputs* $\mathbf{Z}^H = [\mathbf{z}_1^H, \ldots, \mathbf{z}_{M_H}^H]^\top, \mathbf{z}_m^H \in \mathbb{R}^{Q_H}$ with the kernel function $k_H$. Similarly, $\mathbf{K}_{uu}^X$ is computed on another set of *inducing inputs* $\mathbf{Z}^X = [\mathbf{z}_1^X, \ldots, \mathbf{z}_{M_X}^X]^\top, \mathbf{z}_m^X \in \mathbb{R}^{Q_X}$ with the kernel function $k_X$, where $\mathbf{z}_m^X$ has the same dimensionality as the inputs $\mathbf{x}_n$. We construct the conditional distribution of $\mathbf{F}$ as:

$$p(\mathbf{F}|\mathbf{U}, \mathbf{Z}^X, \mathbf{Z}^H, \mathbf{X}, \mathbf{H}) = \mathcal{N}\left(\mathbf{F}_:|\mathbf{K}_{fu}\mathbf{K}_{uu}^{-1}\mathbf{U}_:, \mathbf{K}_{ff} - \mathbf{K}_{fu}\mathbf{K}_{uu}^{-1}\mathbf{K}_{fu}^\top\right), \qquad (4)$$

where $\mathbf{K}_{fu} = \mathbf{K}_{fu}^H \otimes \mathbf{K}_{fu}^X$ and $\mathbf{K}_{ff} = \mathbf{K}_{ff}^H \otimes \mathbf{K}_{ff}^X$. $\mathbf{K}_{fu}^X$ is the cross-covariance computed between $\mathbf{X}$ and $\mathbf{Z}^X$ with $k_X$ and $\mathbf{K}_{fu}^H$ is the cross-covariance computed between $\mathbf{H}$ and $\mathbf{Z}^H$ with $k_H$. $\mathbf{K}_{ff}$ is the covariance matrix computed on $\mathbf{X}$ with $k_X$ and $\mathbf{K}_{ff}^H$ is computed on $\mathbf{H}$ with $k_H$. Note that the prior distribution of $\mathbf{F}$ after marginalizing $\mathbf{U}$ is not changed with the augmentation, because $p(\mathbf{F}|\mathbf{X}, \mathbf{H}) = \int p(\mathbf{F}|\mathbf{U}, \mathbf{Z}^X, \mathbf{Z}^H, \mathbf{X}, \mathbf{H})p(\mathbf{U}|\mathbf{Z}^X, \mathbf{Z}^H)\mathrm{d}\mathbf{U}$. Assuming variational posteriors $q(\mathbf{F}|\mathbf{U}) = p(\mathbf{F}|\mathbf{U}, \mathbf{X}, \mathbf{H})$ and $q(\mathbf{H})$, the lower bound of the log marginal likelihood can be derived as

$$\log p(\mathbf{Y}|\mathbf{X}) \geq \mathcal{F} - \mathrm{KL}\left(q(\mathbf{U}) \,\|\, p(\mathbf{U})\right) - \mathrm{KL}\left(q(\mathbf{H}) \,\|\, p(\mathbf{H})\right), \qquad (5)$$

where $\mathcal{F} = \langle\log p(\mathbf{Y}_:|\mathbf{F}_:)\rangle_{p(\mathbf{F}|\mathbf{U},\mathbf{X},\mathbf{H})q(\mathbf{U})q(\mathbf{H})}$. It is known that the optimal posterior distribution of $q(\mathbf{U})$ is a Gaussian distribution [Titsias, 2009, Matthews et al., 2016]. With an explicit Gaussian definition of $q(\mathbf{U}) = \mathcal{N}\left(\mathbf{U}|\mathbf{M}, \mathbf{\Sigma}^U\right)$, the integral in $\mathcal{F}$ has a closed-form solution:

$$\mathcal{F} = -\frac{ND}{2}\log 2\pi\sigma^2 - \frac{1}{2\sigma^2}\mathbf{Y}_:^\top\mathbf{Y}_: - \frac{1}{2\sigma^2}\mathrm{Tr}\left(\mathbf{K}_{uu}^{-1}\Phi\mathbf{K}_{uu}^{-1}(\mathbf{M}_:\mathbf{M}_:^\top + \mathbf{\Sigma}^U)\right)$$
$$+ \frac{1}{\sigma^2}\mathbf{Y}_:^\top\Psi\mathbf{K}_{uu}^{-1}\mathbf{M}_: - \frac{1}{2\sigma^2}\left(\psi - \mathrm{tr}\left(\mathbf{K}_{uu}^{-1}\Phi\right)\right), \qquad (6)$$

where $\psi = \langle\mathrm{tr}\left(\mathbf{K}_{ff}\right)\rangle_{q(\mathbf{H})}$, $\Psi = \langle\mathbf{K}_{fu}\rangle_{q(\mathbf{H})}$ and $\Phi = \left\langle\mathbf{K}_{fu}^\top\mathbf{K}_{fu}\right\rangle_{q(\mathbf{H})}$.[5] Note that the optimal variational posterior of $q(\mathbf{U})$ with respect to the lower bound can be computed in closed-form. However, the computational complexity of the closed-form solution is $O(NDM_X^2M_H^2)$.

## 3.1  More Efficient Formulation

Note that the lower bound in (5-6) does not take advantage of the Kronecker product decomposition. The computational efficiency could be improved by avoiding directly computing the Kronecker product of the covariance matrices. Firstly, we reformulate the expectations of the covariance matrices $\psi$, $\Psi$ and $\Phi$, so that the expectation computation can be decomposed,

$$\psi = \psi^H\mathrm{tr}\left(\mathbf{K}_{ff}^X\right), \quad \Psi = \Psi^H \otimes \mathbf{K}_{fu}^X, \quad \Phi = \Phi^H \otimes \left((\mathbf{K}_{fu}^X)^\top\mathbf{K}_{fu}^X\right), \qquad (7)$$

where $\psi^H = \left\langle\mathrm{tr}\left(\mathbf{K}_{ff}^H\right)\right\rangle_{q(\mathbf{H})}$, $\Psi^H = \left\langle\mathbf{K}_{fu}^H\right\rangle_{q(\mathbf{H})}$ and $\Phi^H = \left\langle(\mathbf{K}_{fu}^H)^\top\mathbf{K}_{fu}^H\right\rangle_{q(\mathbf{H})}$. Secondly, we assume a Kronecker product decomposition of the covariance matrix of $q(\mathbf{U})$, i.e., $\Sigma^U = \Sigma^H \otimes \Sigma^X$. Although this decomposition restricts the covariance matrix representation, it dramatically reduces

the number of variational parameters in the covariance matrix from $M_X^2 M_H^2$ to $M_X^2 + M_H^2$. Thanks to the above decomposition, the lower bound can be rearranged to speed up the computation,

$$
\begin{aligned}
\mathcal{F} = & -\frac{ND}{2} \log 2\pi\sigma^2 - \frac{1}{2\sigma^2} \mathbf{Y}_:^\top \mathbf{Y}_: \\
& - \frac{1}{2\sigma^2} \operatorname{tr}\left( \mathbf{M}^\top ((\mathbf{K}_{uu}^X)^{-1} \Phi^C (\mathbf{K}_{uu}^X)^{-1}) \mathbf{M} (\mathbf{K}_{uu}^H)^{-1} \Phi^H (\mathbf{K}_{uu}^H)^{-1} \right) \\
& - \frac{1}{2\sigma^2} \operatorname{tr}\left( (\mathbf{K}_{uu}^H)^{-1} \Phi^H (\mathbf{K}_{uu}^H)^{-1} \Sigma^H \right) \operatorname{tr}\left( (\mathbf{K}_{uu}^X)^{-1} \Phi^X (\mathbf{K}_{uu}^X)^{-1} \Sigma^X \right) \\
& + \frac{1}{\sigma^2} \mathbf{Y}_:^\top \left( (\Psi^X (\mathbf{K}_{uu}^X)^{-1}) \mathbf{M} (\mathbf{K}_{uu}^H)^{-1} (\Psi^H)^\top \right)_: - \frac{1}{2\sigma^2} \psi \\
& + \frac{1}{2\sigma^2} \operatorname{tr}\left( (\mathbf{K}_{uu}^H)^{-1} \Phi^H \right) \operatorname{tr}\left( (\mathbf{K}_{uu}^X)^{-1} \Phi^X \right).
\end{aligned}
\tag{8}
$$

Similarly, the KL-divergence between $q(\mathbf{U})$ and $p(\mathbf{U})$ can also take advantage of the above decomposition:

$$
\begin{aligned}
\operatorname{KL}\left( q(\mathbf{U}) \,\|\, p(\mathbf{U}) \right) = & \frac{1}{2} \Bigg( M_X \log \frac{|\mathbf{K}_{uu}^H|}{|\Sigma^H|} + M_H \log \frac{|\mathbf{K}_{uu}^X|}{|\Sigma^X|} + \operatorname{tr}\left( \mathbf{M}^\top (\mathbf{K}_{uu}^X)^{-1} \mathbf{M} (\mathbf{K}_{uu}^H)^{-1} \right) \\
& + \operatorname{tr}\left( (\mathbf{K}_{uu}^H)^{-1} \Sigma^H \right) \operatorname{tr}\left( (\mathbf{K}_{uu}^X)^{-1} \Sigma^X \right) - M_H M_X \Bigg).
\end{aligned}
\tag{9}
$$

As shown in the above equations, the direct computation of Kronecker products is completely avoided. Therefore, the computational complexity of the lower bound is reduced to $O(\max(N, M_H) \max(D, M_X) \max(M_X, M_H))$, which is comparable to the complexity of sparse GPs with independent observations $O(NM \max(D, M))$. The new formulation is significantly more efficient than the formulation described in the previous section. This enables LVMOGP to be applicable to real world scenarios. It is also straight-forward to extend this lower bound to mini-batch learning like in Hensman et al. [2013], which allows further scaling up.

## 3.2 Prediction

After estimating the model parameters and variational posterior distributions, the trained model is typically used to make predictions. In our model, a prediction can be about a new input $\mathbf{x}^*$ as well as a new scenario which corresponds to a new value of the hidden variable $\mathbf{h}^*$. Given both a set of new inputs $\mathbf{X}^*$ with a set of new scenarios $\mathbf{H}^*$, the prediction of noiseless observation $\mathbf{F}^*$ can be computed in closed-form,

$$
\begin{aligned}
q(\mathbf{F}_:^* | \mathbf{X}^*, \mathbf{H}^*) & = \int p(\mathbf{F}_:^* | \mathbf{U}_:, \mathbf{X}^*, \mathbf{H}^*) q(\mathbf{U}_:) \mathrm{d}\mathbf{U}_: \\
& = \mathcal{N}\left( \mathbf{F}_:^* | \mathbf{K}_{f^*u} \mathbf{K}_{uu}^{-1} \mathbf{M}_:, \mathbf{K}_{f^*f^*} - \mathbf{K}_{f^*u} \mathbf{K}_{uu}^{-1} \mathbf{K}_{f^*u}^\top + \mathbf{K}_{f^*u} \mathbf{K}_{uu}^{-1} \Sigma^U \mathbf{K}_{uu}^{-1} \mathbf{K}_{f^*u}^\top \right),
\end{aligned}
$$

where $\mathbf{K}_{f^*f^*} = \mathbf{K}_{f^*f^*}^H \otimes \mathbf{K}_{f^*f^*}^X$ and $\mathbf{K}_{f^*u} = \mathbf{K}_{f^*u}^H \otimes \mathbf{K}_{f^*u}^X$. $\mathbf{K}_{f^*f^*}^H$ and $\mathbf{K}_{f^*u}^H$ are the covariance matrices computed on $\mathbf{H}^*$ and the cross-covariance matrix computed between $\mathbf{H}^*$ and $\mathbf{Z}^H$. Similarly, $\mathbf{K}_{f^*f^*}^X$ and $\mathbf{K}_{f^*u}^X$ are the covariance matrices computed on $\mathbf{X}^*$ and the cross-covariance matrix computed between $\mathbf{X}^*$ and $\mathbf{Z}^X$. For a regression problem, we are often more interested in predicting for the existing condition from the training data. As the posterior distributions of the existing conditions have already been estimated as $q(\mathbf{H})$, we can approximate the prediction by integrating the above prediction equation with $q(\mathbf{H})$, $q(\mathbf{F}_:^* | \mathbf{X}^*) = \int q(\mathbf{F}_:^* | \mathbf{X}^*, \mathbf{H}) q(\mathbf{H}) \mathrm{d}\mathbf{H}$. The above integration is intractable, however, as suggested by Titsias and Lawrence [2010], the first and second moment of $\mathbf{F}_:^*$ under $q(\mathbf{F}_:^* | \mathbf{X}^*)$ can be computed in closed-form.

## 4 Missing Data

The model described in Section 2.2 assumes that for $N$ different inputs, we observe them in all the $D$ different conditions. However, in real world problems, we often collect data at a different set of inputs for each scenario, i.e., for each condition $d$, $d = 1, \ldots, D$. Alternatively, we can view the problem as having a large set of inputs and for each condition only the outputs associated with a

subset of the inputs being observed. We refer to this problem as missing data. For the condition $d$, we denote the inputs as $\mathbf{X}^{(d)} = [\mathbf{x}_1^{(d)}, \ldots, \mathbf{x}_{N_d}^{(d)}]^\top$ and the outputs as $\mathbf{Y}_d = [y_{1d}, \ldots, y_{N_d d}]^\top$, and optionally a different noise variance as $\sigma_d^2$. The proposed model can be extended to handle this case by reformulating the $\mathcal{F}$ as

$$\mathcal{F} = \sum_{d=1}^{D} -\frac{N_d}{2} \log 2\pi\sigma_d^2 - \frac{1}{2\sigma_d^2} \mathbf{Y}_d^\top \mathbf{Y}_d - \frac{1}{2\sigma_d^2} \mathrm{Tr}\left( \mathbf{K}_{uu}^{-1} \Phi_d \mathbf{K}_{uu}^{-1} (\mathbf{M}_:\mathbf{M}_:^\top + \mathbf{\Sigma}^U) \right)$$

$$+ \frac{1}{\sigma_d^2} \mathbf{Y}_d^\top \Psi_d \mathbf{K}_{uu}^{-1} \mathbf{M}_: - \frac{1}{2\sigma_d^2} \left( \psi_d - \mathrm{tr}\left( \mathbf{K}_{uu}^{-1} \Phi_d \right) \right), \tag{10}$$

where $\Phi_d = \Phi_d^H \otimes \left( (\mathbf{K}_{f_d u}^X)^\top \mathbf{K}_{f_d u}^X \right)$, $\Psi_d = \Psi_d^H \otimes \mathbf{K}_{f_d u}^X$, $\psi_d = \psi_d^H \otimes \mathrm{tr}\left( \mathbf{K}_{f_d f_d}^X \right)$, in which $\Phi_d^H = \left\langle (\mathbf{K}_{f_d u}^H)^\top \mathbf{K}_{f_d u}^H \right\rangle_{q(\mathbf{h}_d)}$, $\Psi_d^H = \left\langle \mathbf{K}_{f_d u}^H \right\rangle_{q(\mathbf{h}_d)}$ and $\psi_d^H = \left\langle \mathrm{tr}\left( \mathbf{K}_{f_d f_d}^H \right) \right\rangle_{q(\mathbf{h}_d)}$. The rest of the lower bound remains unchanged because it does not depend on the inputs and outputs. Note that, although it looks very similar to the bound in Section 3, the above lower bound is computationally more expensive, because it involves the computation of a different set of $\Phi_d$, $\Psi_d$, $\psi_d$ and the corresponding part of the lower bound for each condition.

## 5    Related works

LVMOGP can be viewed as an extension of a multiple output Gaussian process. Multiple output Gaussian processes have been thoughtfully studied in Álvarez et al. [2012]. LVMOGP can be seen as an intrinsic model of coregionalization [Goovaerts, 1997] or a multi-task Gaussian process [Bonilla et al., 2008], if the coregionalization matrix $\mathbf{B}$ is replaced by the kernel $\mathbf{K}^H$. By replacing the coregionalization matrix with a kernel matrix, we endow the multiple output GP with the ability to predict new outputs or tasks at test time, which is not possible if a finite matrix $\mathbf{B}$ is used at training time. Also, by using a model for the coregionalization matrix in the form of a kernel function, we reduce the number of hyperparameters necessary to fit the covariance between the different conditions, reducing overfitting when fewer datapoints are available for training. Replacing the coregionalization matrix by a kernel matrix has also been used in Qian et al. [2008] and more recently by Bussas et al. [2017]. However, these works do not address the computational complexity problem and their models can not scale to large datasets. Furthermore, in our model, the different conditions $\mathbf{h}_d$ are treated as latent variables, which are not observed, as opposed to these two models where we would need to provide observed data to compute $\mathbf{K}^H$.

Computational complexity in multi-output Gaussian processes has also been studied before for convolved multiple output Gaussian processes [Álvarez and Lawrence, 2011] and for the intrinsic model of coregionalization [Stegle et al., 2011]. In Álvarez and Lawrence [2011], the idea of inducing inputs is also used and computational complexity reduces to $O(NDM^2)$, where $M$ refers to a generic number of inducing inputs. In Stegle et al. [2011], the covariances $\mathbf{K}^H$ and $\mathbf{K}^X$ are replaced by their respective eigenvalue decompositions and computational complexity reduces to $O(N^3 + D^3)$. Our method reduces computationally complexity to $O(\max(N, M_H) \max(D, M_X) \max(M_X, M_H))$ when there are no missing data. Notice that if $M_H = M_X = M$, $N > M$ and $D > M$, our method achieves a computational complexity of $O(NDM)$, which is faster than $O(NDM^2)$ in Álvarez and Lawrence [2011]. If $N = D = M_H = M_X$, our method achieves a computational complexity of $O(N^3)$, similar to Stegle et al. [2011]. Nonetheless, the usual case is that $N \gg M_X$, improving the computational complexity over Stegle et al. [2011]. An additional advantage of our method is that it can easily be parallelized using mini-batches like in Hensman et al. [2013]. Note that we have also provided expressions for dealing with missing data, a setup which is very common in our days, but that has not been taken into account in previous formulations.

The idea of modeling latent information about different conditions jointly with the modeling of data points is related to the style and content model by Tenenbaum and Freeman [2000], where they explicitly model the style and content separation as a bilinear model for unsupervised learning.

## 6    Experiments

We evaluate the performance of the proposed model with both synthetic and real data.

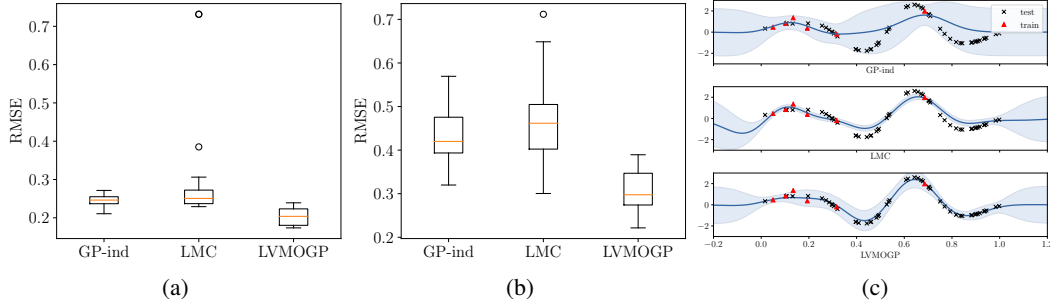

Figure 2: The results on two synthetic datasets. (a) The performance of GP-ind, LMC and LVMOGP evaluated on 20 randomly drawn datasets *without* missing data. (b) The performance evaluated on 20 randomly drawn datasets *with* missing data. (c) A comparison of the estimated functions by the three methods on one of the synthetic datasets with missing data. The plots show the estimated functions for one of the conditions with few training data. The red rectangles are the noisy training data and the black crosses are the test data.

**Synthetic Data.** We compare the performance of the proposed method with GP with independent observations and the linear model of coregionalization (LMC) [Journel and Huijbregts, 1978, Goovaerts, 1997] on synthetic data, where the ground truth is known. We generated synthetic data by sampling from a Gaussian process, as stated in (3), and assuming a two-dimensional space for the different conditions. We first generated a dataset, where all the conditions of a set of inputs are observed. The dataset contains 100 different uniformly sampled input locations (50 for training and 50 for testing), where each corresponds to 40 different conditions. An observation noise with variance 0.3 is added onto the training data. This dataset belongs to the case of no missing data, therefore, we can apply LVMOGP with the inference method presented in Section 3. We assume a 2 dimensional latent space and set $M_H = 30$ and $M_X = 10$. We compare LVMOGP with two other methods: GP with independent output dimensions (GP-ind) and LMC (with a full rank coregionalization matrix). We repeated the experiments on 20 randomly sampled datasets. The results are summarized in Figure 2a. The means and standard deviations of all the methods on 20 repeats are: GP-ind: $0.24 \pm 0.02$, LMC:$0.28 \pm 0.11$, LVMOGP $0.20 \pm 0.02$. Note that, in this case, GP-ind performs quite well because the only gain by modeling different conditions jointly is the reduction of estimation variance from the observation noise.

Then, we generated another dataset following the same setting, but where each condition had a different set of inputs. Often, in real data problems, the number of available data in different conditions is quite uneven. To generate a dataset with uneven numbers of training data in different conditions, we group the conditions into 10 groups. Within each group, the numbers of training data in four conditions are generated through a three-step stick breaking procedure with a uniform prior distribution (200 data points in total). We apply LVMOGP with missing data (Section 4) and compare with GP-ind and LMC. The results are summarized in Figure 2b. The means and standard deviations of all the methods on 20 repeats are: GP-ind: $0.43 \pm 0.06$, LMC:$0.47 \pm 0.09$, LVMOGP $0.30 \pm 0.04$. In both synthetic experiments, LMC does not perform well because of overfitting caused by estimating the full rank coregionalization matrix. The figure 2c shows a comparison of the estimated functions by the three methods for a condition with few training data. Both LMC and LVMOGP can leverage the information from other conditions to make better predictions, while LMC often suffers from overfitting due to the high number of parameters in the coregionalization matrix.

**Servo Data.** We apply our method to a servo modeling problem, in which the task is to predict the rise time of a servomechanism in terms of two (continuous) gain settings and two (discrete) choices of mechanical linkages [Quinlan, 1992]. The two choices of mechanical linkages introduce 25 different conditions in the experiments (five types of motors and five types of lead screws). The data in each condition are scarce, which makes joint modeling necessary (see Figure 3a). We took 70% of the dataset as training data and the rest as test data, and randomly generated 20 partitions. We applied LVMOGP with a two-dimensional latent space with an ARD kernel and used five inducing points for the latent space and 10 inducing points for the function. We compared LVMOGP with GP with ignoring the different conditions (GP-WO), GP with taking each condition as an independent output (GP-ind), GP with one-hot encoding of conditions (GP-OH) and LMC. The means and standard deviations of the RMSE of all the methods on 20 partitions are: GP-WO: $1.03 \pm 0.20$, GP-ind:

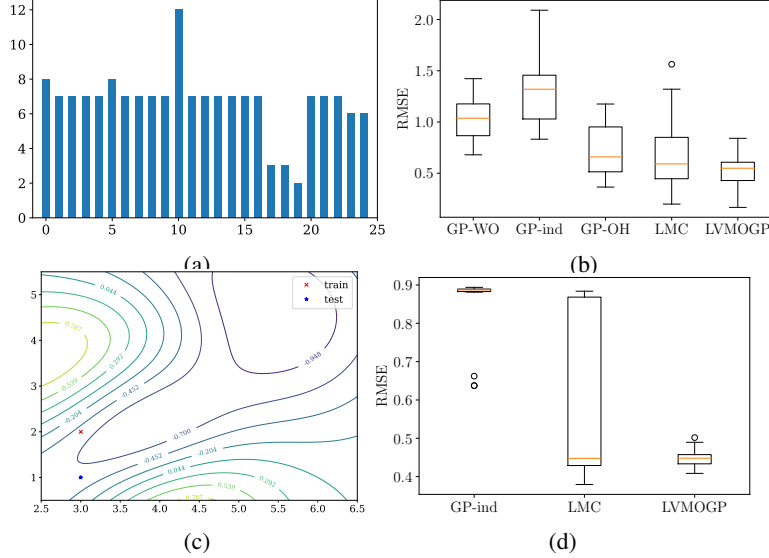

(a)  (b)

(c)  (d)

Figure 3: The experimental results on servo data and sensor imputation. (a) The numbers of data points are scarce in each condition. (b) The performance of a list of methods on 20 different train/test partitions is shown in the box plot. (c) The function learned by LVMOGP for the condition with the smallest amount of data. With only one training data, the method is able to extrapolate a non-linear function due to the joint modeling of all the conditions. (d) The performance of three methods on sensor imputation with 20 repeats.

$1.30 \pm 0.31$, GP-OH: $0.73 \pm 0.26$, LMC:$0.69 \pm 0.35$, LVMOGP $0.52 \pm 0.16$. Note that in some conditions the data are very scarce, e.g., there are only one training data point and one test data point (see Figure 3c). As all the conditions are jointly modeled in LVMOGP, the method is able to extrapolate a non-linear function by only seeing one data point.

**Sensor Imputation.** We apply our method to impute multivariate time series data with massive missing data. We take a in-house multi-sensor recordings including a list of sensor measurements such as temperature, carbon dioxide, humidity, etc. [Zamora-Martínez et al., 2014]. The measurements are recorded every minute for roughly a month and smoothed with 15 minute means. Different measurements are normalized to zero-mean and unit-variance. We mimic the scenario of massive missing data by randomly taking out 95% of the data entries and aim at imputing all the missing values. The performance is measured as RMSE on the imputed values. We apply LVMOGP with missing data with the settings: $Q_H = 2$, $M_H = 10$ and $M_X = 100$. We compare with LMC and GP-ind. The experiments are repeated 20 times with different missing values. The results are shown in a box plot in Figure 3d. The means and standard deviations of all the methods on 20 repeats are: GP-ind: $0.85 \pm 0.09$, LMC:$0.59 \pm 0.21$, LVMOGP $0.45 \pm 0.02$. The high variance of LMC results are due to the large number of parameters in the coregionalization matrix.

## 7  Conclusion

In this work, we study the problem of how to model multiple conditions in supervised learning. Common practices such as one-hot encoding cannot efficiently model the relation among different conditions and are not able to generalize to a new condition at test time. We propose to solve this problem in a principled way, where we learn the latent information of conditions into a latent space. By exploiting the Kronecker product decomposition in the variational posterior, our inference method is able to achieve the same computational complexity as sparse GPs with independent observations, when there are no missing data. In experiments on synthetic and real data, LVMOGP outperforms common approaches such as ignoring condition difference, using one-hot encoding and LMC. In Figure 3b and 3d, LVMOGP delivers more reliable performance than LMC among different train/test partitions due to the marginalization of latent variables.

**Acknowledgements**  MAA has been financed by the Engineering and Physical Research Council (EPSRC) Research Project EP/N014162/1.

## Footnotes

[4]Assume that the number of inducing points is proportional to $D$.

[5]The expectation with respect to a matrix $\langle\cdot\rangle_{q(\mathbf{H})}$ denotes the expectation with respect to every element of the matrix.

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
