[Reviews · NeurIPS 2017]

Reviewer 1



# Update after author feedback and reviewer discussion I thank the authors for feedback. I maintain my assessment and do not recommend publication at this stage. The core contribution is representing conditions with latent variables, and deriving a VI algorithm to cope with intractibility. This is interesting, but the discussion around it could be much improved. Some possible improvements are addressed in the author feedback, eg I'm not sure how Fig 1 could have been understood without the complementary explanation brought up in the feedback. Beyond what has been addressed in the author feedback, some work is needed to make this paper appealing (which the idea under study, the method and the results seem to call for): - clarifying the mathematical formulation, eg what forms of $k_H$ are we examining, provide a full probabilistic model summary of the model, point out design choices - pointing out differences or similarities with existing work - remove gratuitous reference to deep learning in intro (it detracts) - make sure that all important questions a reader might have are addressed # Overall assessment The issue addressed (modelling univariate outputs which were generated under different, known conditions) and the modelling choice (representing conditions as latent variables) are interesting. I see no reason why to even mention and work with the (restrictive) parameterization of section 2.2, and not immediately the parameterization allowing "missing data" section 4. The latter is more interesting and equally simple. Overall, the paper is reasonably clear, but seems lacking in originality and novelty. ## Novelty Wrt work on multi-output / multi-task GP, I can see that: - this model has latent variables, not task IDs, and so can generalize over tasks (called conditions in the paper) - the computational treatment using inducing points helps The two seem very close though. The assumption of Kronecker structure is a staple. In particular, experiments should try compare to these methods (eg considering every condition a distinct task). The application of Titsias 2009 seems straightforward, I cannot see the novelty there, nor why so much space is used to present it. ## Presentation and clarity 1. Paper requires English proofreading. I usually report typos and mistakes in my reviews, but here there are too many grammar errors. 2. Motivating example with braking distance is helpful. 2. I don't understand fig 1 c,d at all: why do we not see the data points of fig 1a? where do these new points come from? what is depicted in the top vs the bottom plot? 2. References: the citation of Bussas 2017 is wrong. The complete correct citation is more like: Matthias Bussas, Christoph Sawade, Tobias Scheffer, and Niels Landwehr. Varying-coefficient models for geospatial transfer learning.. Machine Learning, doi:10.1007/s10994-017-5639-3, 2017.

Reviewer 2



This paper presents a method for inference in GPs with multiple outputs where the outputs are highly correlated. The model is augmented with additional latent variables responsible to model this correlation, which are estimated with variational inference. This is a interesting paper that introduces a (seemingly novel?) method to model the correlation between multiple GPs with a latent variable instead of imposing a certain structure in the kernel (such as the LMC does). The results looks good and comes with great computational complexity. I would prefer a more though discussion investigating why this approach outperforms the standard methods. Also, Section 3 is a bit difficult to follow, where the intuition and the pedagogical skill of the presentation is somewhat lacking. If the presentation could be improved, it would help the accessibility of the paper.

Reviewer 3



The paper extends multi output GP models by modelling the outputs as a function on the data and a shared latent space which aims to find embeddings for each condition. A variational GP approach is taken where the variational distribution is factorised as kronecker matrix Uhh \kron Uff (which seems a reasonable assumption), allowing fast inference. The key contribution of this paper is that by modelling the conditions in the latent space, it is possible to marginalise these out at test time, providing a smooth average across different condition seen at training train, weighted by the similarity to these conditions. While not applicable to the general case of missing data, I would be interested to see how this model performs in the fully observed case compared to multi-output models like Wilson et al.'s Gaussian Process Regression Network. This is just curiosity and doesn't impact my positive review of this paper. Minor errors: line 45: "physics about car"